# Bifunctionality of dirhodium tetracarboxylates in metallaphotocatalysis

Taoda Shi [1,2,3] ✉, Tianyuan Zhang[1,2], Jiying Yang[1], Yukai Li[1], Jirong Shu[1], Jingyu Zhao[1], Mengchu Zhang[1], Dan Zhang[1] & Wenhao Hu [1,3] ✉

Metallaphotocatalysis has been recognized as a pivotal catalysis enabling new reactivities. Traditional metallaphotocatalysis often requires two or more separate catalysts and exhibits flaw in cost and substrate-tolerance, thus representing an await-to-solve issue in catalysis. We herein realize metalla-photocatalysis with a bifunctional dirhodium tetracarboxylate ([Rh$_2$]) alone. The [Rh$_2$] shows an photocatalytic activity of promoting singlet oxygen ($^1$O$_2$) oxidation. By harnessing its photocatalytic activity, the [Rh$_2$] catalyzes a photochemical cascade reaction (PCR) via combination of carbenoid chemistry and $^1$O$_2$ chemistry. The PCR is characterized by high atom-efficiency, excellent stereoselectivities, mild conditions, scalable synthesis, and pharmaceutically interesting products. DFT calculations-aided mechanistic study rationalizes the reaction pathway and interprets the origin of stereo-selectivities of the PCR. The products show inhibitory activity against PTP1B, being promising in the treatment of type II diabetes and cancers. Overall, here we show the bifunctional [Rh$_2$] merges Rh-carbenoid chemistry and $^1$O$_2$ chemistry.

Metallaphotocatalysis, which merges metal catalysis and photo-catalysis, is an important catalytic strategy for organic synthesis and for the expansion of chemical space[1–6]. Traditional metallaphotocatalysis usually requires two or more catalysts and shows limitations in cost and substrate-tolerance. Hence, the development of unimolecular metallaphotocatalysis using bifunctional catalysts will be attractive (Fig. 1A). Glorious[7], Gevorgyan[8], Shang and Fu[9], Bach[10], Meggers[11], Xiao[12], Chen[13] and others[14,15] have developed bifunctional catalysts that were generated from elegant design and remarkable synthesis. Our group has a lasting interest in transitional metal-catalyzed carbenoid-involved reactions[16–19]. Here, we intend to find a bifunctional transitional metal catalyst to connect carbenoid chemistry with photochemistry. The transitional metal complex photosensitizers (PSs) work via different charge transfer modes including metal-to-ligand charge transfer (MLCT) and ligand-to-metal charge transfer, metal-to-metal charge transfer (MMCT) and ligand-to-ligand charge transfer (Fig. 1B, left)[20,21]. MLCT is a major transition mode in photocatalysis[22–24]. By contrast, MMCT is seldom used for photocatalysis[25]. To the best of our knowledge, MMCT in dirhodium complex has not been harnessed for photocatalysis. The known dirhodium PSs often lack activity for carbenoid generation since their axial orbitals have been occupied by strong coordinated ligands, leaving no binding site for carbenoid precursors like diazo compounds (Fig. 1B). Therefore, we decided to search for photocatalytic activity from active carbenoid transfer catalysts. To this end, we performed time-dependent density functional theory (TD-DFT)-based screening on the known carbenoid transfer catalysts and selected the ones with absorption wavelength at the range of visible light (Fig. 1B)[26–28]. Consequently, dirhodium tetra-carboxylates ([Rh$_2$]) were predicted as potential PSs. Then the bifunctional [Rh$_2$] were chosen to develop photochemical cascade reactions (PCR) by merging carbenoid chemistry and $^1$O$_2$ chemistry and to create a new chemical space (Fig.1C). [Rh$_2$] are well known for their versatility in carbenoid[29–31] and nitrene transfer reactions[32–35], and oxidations[36,37] (Fig. 1D, left). However, their utility in photocatalysis is less explored than its neighbors in the periodic table including nickel, ruthenium, and iridium catalysts[38–40]. Meggers's group invented bis-

[1]School of Pharmaceutical Sciences, Sun Yat-sen University, Guangzhou 510006, China. [2]These authors contributed equally: Taoda Shi, Tianyuan Zha. [3]These authors jointly supervised this work: Taoda Shi, Wenhao Hu. ✉e-mail: shitd@mail.sysu.edu.cn; huwh9@mail.sysu.edu.cn

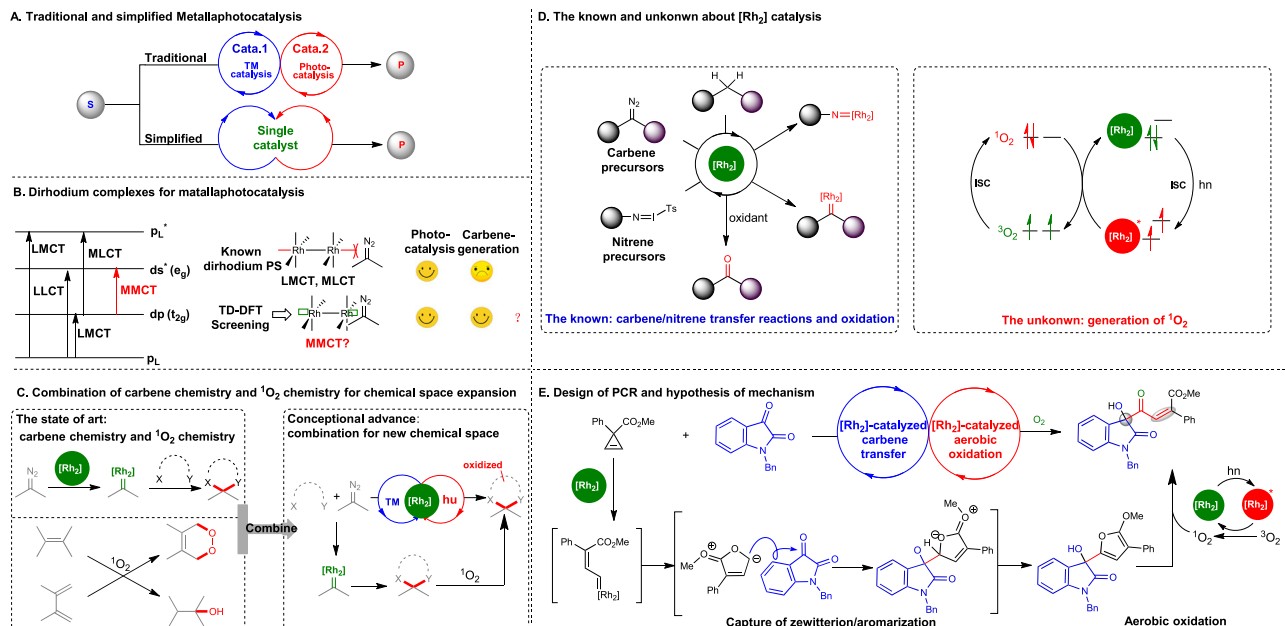

**Fig. 1 | The significance of bifunctionality of [Rh₂] in metallaphotocatalysis.**
**A** Traditional and simplified metallaphotocatalysis. **B** Dirhodium complexes for metallaphotocatalysis. **C** Combination of carbenoid chemistry and singlet oxygen chemistry. **D** The known and unknown about [Rh₂] catalysis. **E** Design of PCR and hypothesis of mechanism. S substrate, P product, TM transitional metal, LLCT ligand-to-ligand charge transfer, LMCT ligand-to-metal-ligand transfer, MLCT metal-to-ligand transfer, MMCT metal-to-metal charge transfer, PS photosensitizer, ISC intersystem crossing.

cyclometalated rhodium complexes functioning as both chiral Lewis acid and PS[41–43], representing the precedence of bifunctional rhodium catalysts in metallaphotocatalysis. Since $^1O_2$ chemistry is widely used in the synthesis of drugs or natural products[44], we thus asked if the frequently used [Rh₂] catalyst in carbenoid transfer reactions could function as a PS generating $^1O_2$ (Fig. 1D, right). In this work, we present a bifunctional [Rh₂]-catalyzed PCR constituted by carbenoid transfer and $^1O_2$ oxidation, synthesizing pharmaceutically interesting 3-acetyl-3-hydroxyl oxindoles (Fig. 1E).

## Results

### Characterization of [Rh₂] as bifunctional catalysts for PCRs

The TD-DFT calculations of Rh₂(OAc)₄ revealed two absorption bands in the visible spectrum (Fig. 2A, 400–500 and 500–900 nm), implicating Rh₂(OAc)₄ was potentially active as a PS. Further molecule orbitals analysis on Rh₂(OAc)₄ indicated that MMCT transition between Rh-Rh π* and Rh-Rh σ* was responsible for the higher absorption band (Fig. 2B, Left, red arrow), and the MLCT transition between Rh-Rh π* and Rh-O σ* contributed to the lower absorption band (Fig. 2B, Left, blue arrow). The pictures of these three orbitals of Rh₂(OAC)₄ were presented (Fig. 2B, right). Therefore, Rh₂(OAc)₄ as a potential photocatalyst differentiated itself from the conventional MLCT photocatalysts by taking both MLCT and MMCT transition modes. To validate the computational results, we collected experimental UV-vis spectrum of Rh₂(OAc)₄ and Rh₂(esp)₂ in the solvents including ethyl acetate (EtOAc), acetonitrile, methanol, tetrahydrofuran (THF), acetone, and toluene. As shown in Fig. 2C, D, both dirhodiums produced two absorption bands in the visible spectrum and gave relatively strong absorption between 500 and 900 nm. With this information, we anticipated the dirhodium catalysts to be used in visible light catalysis. To figure out whether the MMCT transition contributed to the photocatalytic activity, 5 W, 640–650 nm red light was used and the diphenyl benzofuran (DPBF) 1 could be smoothly oxidized. Meanwhile, 5 W, 440–450 nm blue light was evaluated and oxidation of DPBF 1 finished quickly as well (Supplementary Fig. 10). The evidence demonstrated the [Rh₂] catalyzed the photocatalytic oxidation by adopting both MMCT and MLCT transition modes.

Considering the harmlessness of white light, the white LED was adopted in the following experiments. Next, we designed a panel of oxidation experiments by using a probe of singlet oxygen, DPBF 1 as a substrate. DPBF 1 could be oxidized to diketone 3 under the conditions of using [Rh₂] as PSs, 12 W white LED as light source, air as oxidant and CDCl₃ as solvent (Supplementary Fig. 11, condition a and b). Deprivation of [Rh₂] catalyst (Supplementary Fig. 11, condition c), white light (Supplementary Fig. 11, condition d), or oxygen (Supplementary Fig. 11, condition e) lead to the inhibition of the oxidation (quantitative data were summarized in the Supplementary Table 1). Conclusively, Rh₂(OAc)₄ and Rh₂(esp)₂ are able to catalyze the generation of $^1O_2$ and hold potential in developing PCRs by merging carbenoid chemistry and $^1O_2$ chemistry.

### Optimization of condition

With the photocatalytic activity of Rh₂(OAc)₄ and Rh₂(esp)₂ being confirmed, we set out to test them as the catalyst of PCR of isatin 4a and cyclopropene carboxylate 5a. Interestingly, the reaction went smoothly with Rh₂(esp)₂ as a single catalyst, producing the desired product 6a with >20:1 Z/E ratio. Then a group of Brønsted acids, including *rac*-BNDHP, HOAc, PhCO₂H, and *p*-TSA, were evaluated as a cocatalyst, and carboxylic acids were proved to be better, elevating about 10% yield (Table 1, entry 2 and 3 vs. entry 1), while phosphoric acid and sulfonic acid resulted in no desired product (Table 1, entry 4 and 5). The inhibition of generating 6a was possibly derived from the strong acidity of the two acids, which increased the hydrolysis rate of intermediate 7a (Supplementary Table 2). Lowering the loading of HOAc to 0.1 eq. decreased the yield by 13% (Table 1, entry 6 vs. entry 2). Then a series of the [Rh₂] including Rh₂(OAc)₄, Rh₂(Oct)₄, Rh₂(cap)₄, and Rh₂(TFA)₄ were tested. Rh₂(OAc)₄ and Rh₂(Oct)₄ showed comparable efficiency with Rh₂(esp)₂ (Table 1, entry 7 and 8 vs. entry 2), but the catalytic reactivity of Rh₂(cap)₄, and Rh₂(TFA)₄ drastically declined (Table 1, entry 9 and 10). Then the temperature parameter was adjusted, and 35 °C was optimal with 8% yield increase (Table 1, entry 11 vs. entry 2). However, a further increase in temperature gained no benefit (Table 1, entry 12). The deprivation of oxygen completely inhibited the generation of the desired product 6a and gave intermediate 7a

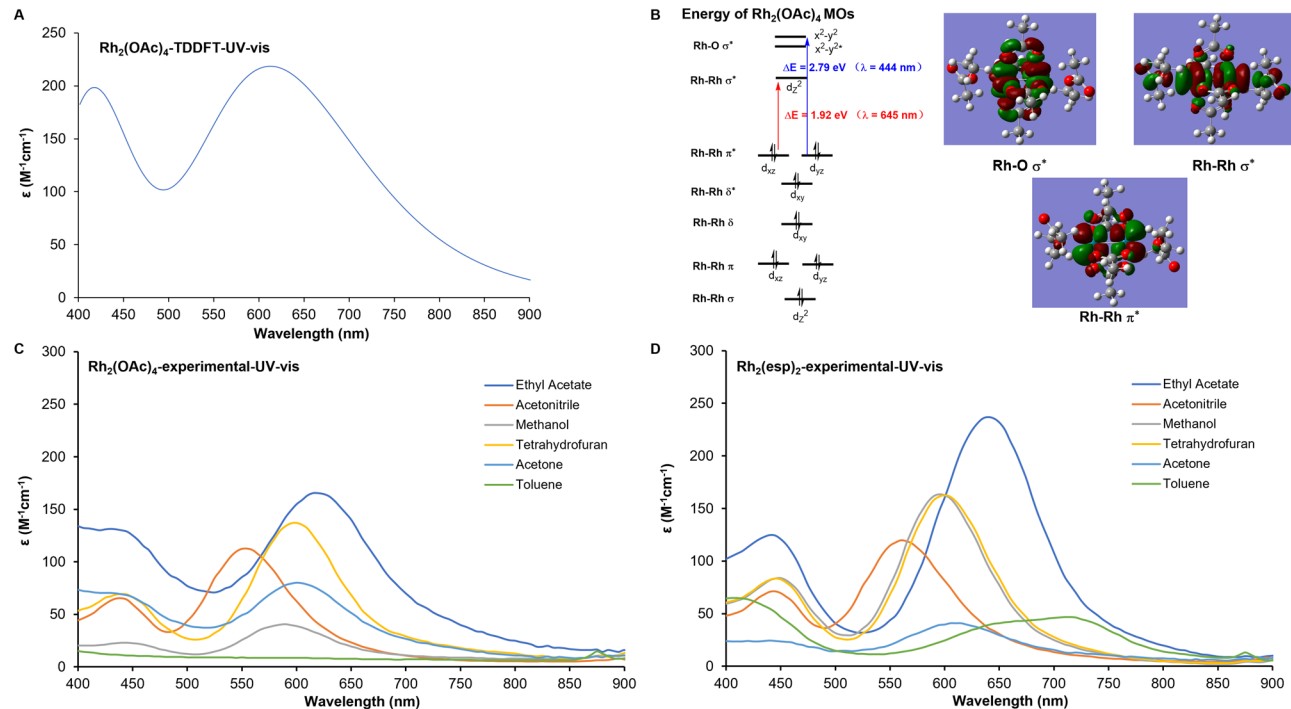

**Fig. 2 | UV-vis profile of Rh₂(OAc)₄ and Rh₂(esp)₂. A** TD-DFT-generated UV-vis spectrum of Rh₂(OAc)₄. **B** Molecule orbitals (MOs) analysis of Rh₂(OAc)₄. **C** Experimental UV-vis spectrum of Rh₂(OAc)₄. **D** Experimental UV-vis spectrum of Rh₂(esp)₂.

(Table 1, entry 13). However, the effect of pure oxygen was equal to the effect of air (Table 1, entry 14 vs. entry 12). The decrease in the amount of **5a** from 1.5 eq. to 1.2 eq. reduced the yield by 9% (Table 1, entry 15 vs. entry 12). There was no significant difference between sunlight and LED in the reaction (Table 1, entry 16 vs. entry 12). The influence of reaction parameters on the yield was concluded in the radar diagram. Rh₂(esp)₂, oxygen, and visible light are essential to the reaction, supporting our initial mechanism hypothesis. Eventually, the standard conditions were established as "2 mol% Rh₂(esp)₂, 1.5 eq. HOAc, 1.5 eq. **5a**, EtOAc as solvent, 35 °C, 12 W white LED light as light source, air as oxidant".

## Substrate scope

We then started to investigate the substrate scope of the PCRs. Firstly, we evaluated the effect of a panel of R¹ groups containing benzyl (Bn), hydrogen, *tert*-butoxycarbonyl (Boc), acetyl (Ac) and methyl (Me) groups. Benzyl group gave the highest yield (**6a**, 81%), with hydrogen giving the second highest yield (Fig. 3, **6b**, 76%). Boc, Ac, and methyl groups provided similar yields (Fig. 3, **6c**, 71%; **6d**, 70% and **6e**, 68%). Then Bn for R¹ group was chosen to investigate the influence of R² groups. The 4 positions at the left ring of isatin could be substituted without affecting the reactivity. Specifically, 4-Cl, 5-Cl, 6-Cl or 7-Cl-substituted istains produced the desired product with 68–78% yields (Fig. 3, **6f**–**i**). Both electron-withdrawing groups (EWGs) and electron-donating groups (EDGs) at 5-position were tolerated (Fig. 3, **6j**, 82%; **6k**, 80%; and **6l**, 78%). 6-MeO-substituted isatin also afforded good yield (Fig. 3, **6m**, 77%). Then isatin **4a** was selected as a standard substrate to investigate the scope of cyclopropene carboxylates. The halos, EDG and EWG groups on the *para*-position Ar group all resulted in good yields (Fig. 3, **6n**, **6q**–**s**, 77–81% yield). *Ortho*- and *meta*-substitutions were tolerated as well (Fig. 3, **6o**–**p**, 80–82% yields). Furthermore, 3,4,5-(MeO)₃-substitution is compatible, giving 76% yield (Fig. 3, **6t**). The cross-match of *N*-Me isatin and 3,4,5-(MeO)₃-substituted cyclopropene generated **6u** in 75% yield. Ethyl cyclopropene ester gave the desired product **6v** in 79% yield. In a word, the PCR showed broad substrate scope. Gladly, compound **6a** provided single

crystals with good enough quality to collect crystallography data. The X-ray structure is displayed to confirm the configuration of the double C-C bond of **6a** as Z. In terms of the stereoselectivities, all the products were all harvested with the ratio of Z-isomer and E-isomer being >20:1, demonstrating excellent stereo control in the reaction. Overall, the PCR, efficiently constructing multiple types of bonds and one quaternary center under mild conditions, represents one of the sustainable reactions. It should be noted that the motif 1,4-dicarbonyl Z-alkenes frequently exists in bioactive compounds[45,46]. However, the straightforward access to the motif remains limited by the fact that most of the construction methods of C-C double bond are E-selective and only one recently published example is Z-selective (Fig. 3)[47]. In addition, the 3-hydoxy oxindole skeleton is widely presented in bioactive compounds and natural products and is important for drug discovery[48]. In light of the biological interest of product **6**, which is inaccessible otherwise, an *in virtual* library of product **6** was created via rdkit (open-source cheminformatics tool available at: https://rdkit. org/) based on the reactivity scope of the PCR to promote further drug discovery campaign (the library is available at http://www.sysu-sps-compound.com/ui/#/molecule-library/29).

## Gram-scale reaction and synthetic utility

A gram-scale reaction was performed and 1.37 g of desired product **6a** was isolated in 80% yield as a white solid after simple filtration. The convenience of large-scale reaction allowed us to quickly explore synthetic utility of product **6a**. As shown in Fig. 4, compound **6a** could be transformed to 1,6-dihydropyridazine derivative **7** in 98% yield with the treatment of hydrazine hydrate. When **6a** was treated with NaBH₄, a γ-lactone **8** was obtained as a single stereoisomer in 95% yield, with ketone carbonyl group and C-C double bond being reduced simultaneously, while only C-C double bond was reduced when palladium/charcoal-catalyzed hydrogenation was applied to compound **6a**, giving compound **9** in 98% yield and 64:36 dr. The compound **6a** bearing α, β-unsaturated ketonester moiety was thought to undergo interesting transformations under basic conditions. Intriguingly, compound **6a** underwent a rearrangement to form spirooxindole **10a** containing 3

**Table 1 | Condition screening of the [Rh₂]-enabled carbenoid transfer-aerobic oxidation cascade reaction[a]**

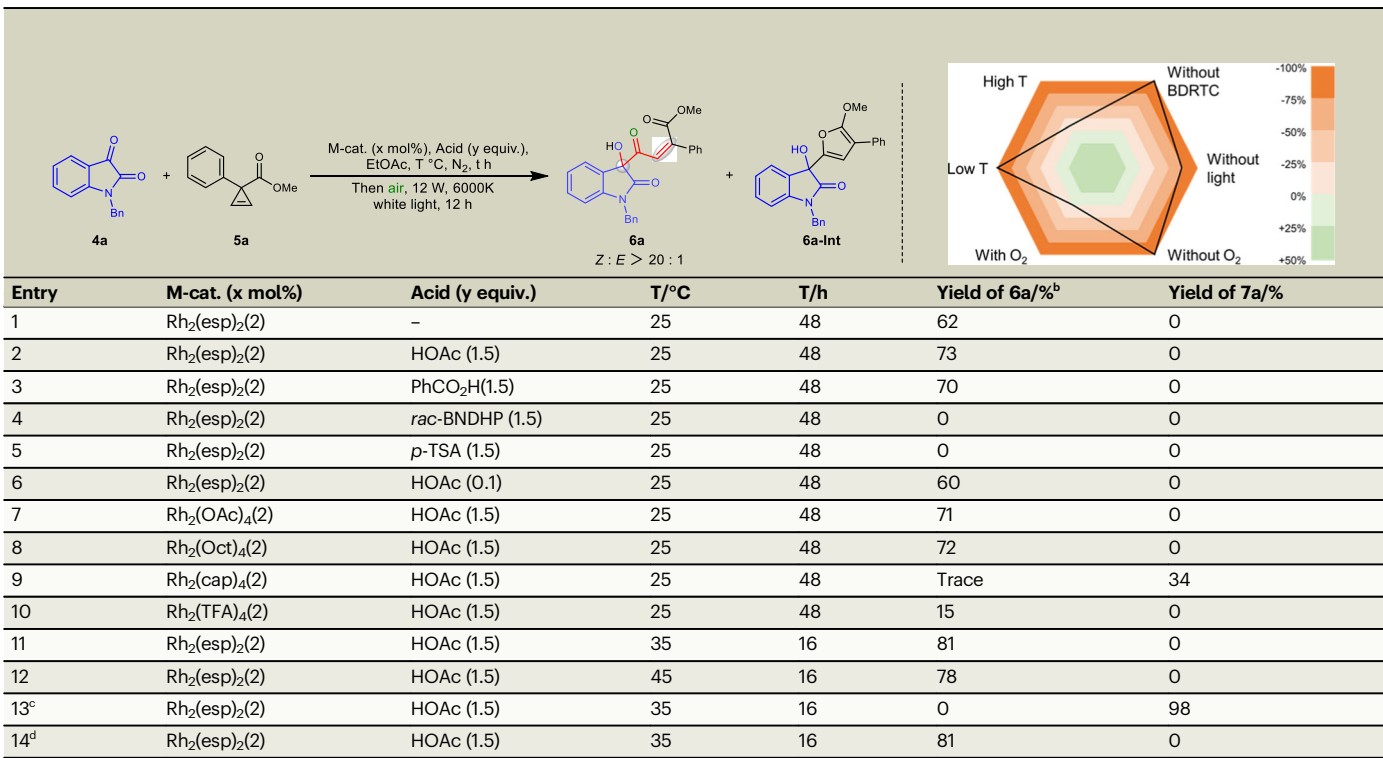

| Entry | M-cat. (x mol%) | Acid (y equiv.) | T/°C | T/h | Yield of 6a/%[b] | Yield of 7a/% |
|---|---|---|---|---|---|---|
| 1 | Rh₂(esp)₂(2) | – | 25 | 48 | 62 | 0 |
| 2 | Rh₂(esp)₂(2) | HOAc (1.5) | 25 | 48 | 73 | 0 |
| 3 | Rh₂(esp)₂(2) | PhCO₂H(1.5) | 25 | 48 | 70 | 0 |
| 4 | Rh₂(esp)₂(2) | rac-BNDHP (1.5) | 25 | 48 | 0 | 0 |
| 5 | Rh₂(esp)₂(2) | p-TSA (1.5) | 25 | 48 | 0 | 0 |
| 6 | Rh₂(esp)₂(2) | HOAc (0.1) | 25 | 48 | 60 | 0 |
| 7 | Rh₂(OAc)₄(2) | HOAc (1.5) | 25 | 48 | 71 | 0 |
| 8 | Rh₂(Oct)₄(2) | HOAc (1.5) | 25 | 48 | 72 | 0 |
| 9 | Rh₂(cap)₄(2) | HOAc (1.5) | 25 | 48 | Trace | 34 |
| 10 | Rh₂(TFA)₄(2) | HOAc (1.5) | 25 | 48 | 15 | 0 |
| 11 | Rh₂(esp)₂(2) | HOAc (1.5) | 35 | 16 | 81 | 0 |
| 12 | Rh₂(esp)₂(2) | HOAc (1.5) | 45 | 16 | 78 | 0 |
| 13[c] | Rh₂(esp)₂(2) | HOAc (1.5) | 35 | 16 | 0 | 98 |
| 14[d] | Rh₂(esp)₂(2) | HOAc (1.5) | 35 | 16 | 81 | 0 |
| 15[e] | Rh₂(esp)₂(2) | HOAc (1.5) | 35 | 16 | 72 | 0 |
| 16[f] | Rh₂(esp)₂(2) | HOAc (1.5) | 35 | 16 | 80 | 0 |

[a]Unless otherwise indicated, reaction conditions: **4a** (0.20 mmol), **5a** (0.30 mmol), metal catalyst, solvent (2 mL), 12 W white LED. Rh2(oct)4: Dirhodium(II) octanoate; Rh2(cap)4: Dirhodium(II) Tetra(caprolactam); Rh2(TFA)4: Dirhodium(II) trifluoroacetate.
[b]Isolated yield.
[c]Under an atmosphere of nitrogen instead of air.
[d]Under an atmosphere of pure oxygen instead of air.
[e]0.24 mmol **5af**.
[f]In sunlight instead of LED, 8 h. In the radar diagram: T temperature.

stereocenters in 71% yield and >95:5 dr. The interesting finding motivated us to perform the PCR and rearrangement transformations in a sequential process. Gladly, we could obtain the desired product **10a** in moderate yield (Fig. 5). Then we further evaluated the substrate scope of this protocol. As shown in Fig. 5, EDG, EWG and halos could be tolerated and 11 spirooxindoles could be synthesized in 35–60% yields and 86:14– >95:5 dr. The sequential protocol forms multiple bonds and three sequential chiral centers, representing an alternative efficient construction of a biologically interesting γ-lactone spirooxindole scaffold[48]. The rearrangement of the 5-hydoxy-1,4-dicarbonyl Z-alkene is featured by high chemoselectivity, high diastereoselectivity, convenient handling in water and air, and easy purification, representing an ideal reaction for green synthesis. In addition, we intended to conduct a strictly one-pot PCR/hydrolysis/cyclization cascade reaction. Interestingly, the intermediate **12** was isolated, which could convert into the final product **10** (Fig. 5, top). The result indicated a one-pot procedure to compound **12** is feasible. Lastly, an *in virtual* library of the γ-lactone spirooxindoles was built as well for the coming bioactivity screening and the data are free and accessible at the aforementioned website.

## Mechanism study

Firstly, we postulated that compound **6a-Int** was a key intermediate that was oxidized by singlet oxygen to produce final product **6a**. To prove the speculation, we used pure **6a-Int** to perform oxidation under standard conditions and provided **6a** in 83% yield, implicating **6a** was oxidized from **6a-Int** (Supplementary Fig. 12, Eq. 1 condition A).

Eliminating anyone of the conditions including visible light, Rh₂(esp)₂ and oxygen caused a significant reduction in the yield (Supplementary Fig. 12, Eq. 1, condition B, C, and D). Secondly, to figure out the generation of intermediate **6a-Int** is through Friedel–Crafts reaction of furan **11** with isatin **4a** or through the capture of transient zwitterion by **4a**, we generated furan **11** from **5a** in situ and added **4a** subsequently. The reaction gave no intermediate **6a-Int** (Supplementary Fig. 12, Eq. 2), excluding the pathway of the Friedel–Crafts reaction pathway. Lastly, we asked whether the zwitterion is dirhodium-associated or free during the capture process. To answer the question, we tested a group of four chiral dirhodium catalysts including Rh₂(S-PTTL)₄, Rh₂(S-PTPA)₄, Rh₂(R-BPTTL)₄, and Rh₂(R-BTPCB)₄, and only detected racemic product **6a-Int** (Supplementary Fig. 12, Eq. 3), indicating [Rh₂] catalyst is not associated with the zwitterion when reacting with **4a**.

Based on the observation of control experiments described above, we characterized the mechanism of the PCR via DFT calculations. The computational study was trying to answer the following questions: (1) Whether the dirhodium-associated zwitterion or the free zwitterion participated in the capture process? (2) What is the function of HOAc in the cascade reaction? (3) What is the origin of Z-selectivity of product **6a**? (4) How to control diastereoselectivities in LiOH-promoted rearrangement? First of all, as shown in Gibbs energy map (Fig. 6A), the ΔG$_{sol}$ of the free zwitterion **Int3** was 1.21 kcal/mol lower than the one of the dirhodium-associated zwitterion **Int2**, implicating **Int3** was the more preferential intimate for the coming capture process. The assumption was supported by the results that only racemic **6a** was produced in the asymmetric catalysis using chiral [Rh₂]

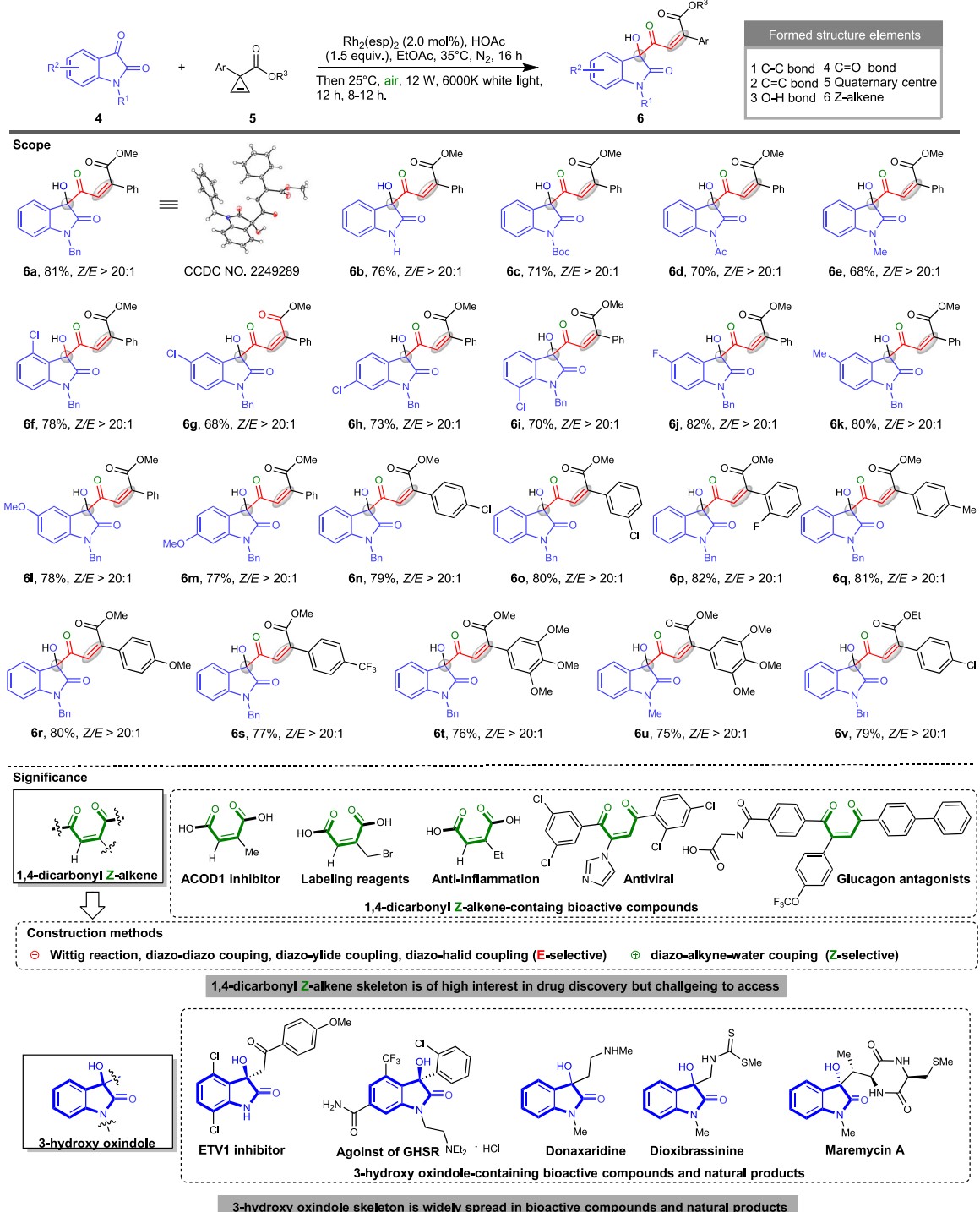

**Fig. 3 | Substrate scope and significance of Rh₂(esp)₂-catalyzed PCR of isatins and cyclopropenes.** Reaction conditions: **4a** (0.20 mmol), **5a** (0.30 mmol), Rh₂(esp)₂ (2.0 mol%), ethyl acetate (2 mL), sunlight.

(Supplementary Fig. 12, Eq. 3), further indicating the chiral [Rh₂] catalysts were unbonded in the active intermediate. Upon the role of HOAc, we considered it functions as a bifunctional Brønsted acid in the process of trapping zwitterion **Int3** by isatin and as a reductant in the process of oxidation of **7a** by ¹O₂. In **TS2**, HOAc activated isatin via H-bonding interaction (1.43 Å) to facilitate the nucleophilic addition to isatin (1.73 Å). On the other hand, HOAc as a Brønsted base grabbed the proton of the zwitterion **Int3** (2.01 Å) to promote the aromatization. Interestingly, if HOAc was removed from the reaction system, the capture process could go through [3+2] cycloaddition between

zwitterion **Int3** and isatin (**TS3**). However, we failed to isolate and characterize the presumed intermediate due to its short half-life. We still could not exclude the pathway because **TS3** had a 9.93 kcal/mol lower energy barrier than **TS2**. In the cycle of singlet oxygen oxidation, HOAc was thought to open the trioxygene-containing five-membered ring of **Int4** via nucleophilic attack to form **Int5**. Subsequently, the removal of HOOAc initiated by the intramolecular nucleophilic attack at acetyl group gave the final product **Z-6**. Thirdly, the origin of stereoselectivity toward Z-isomer was summarized in the following: (i) the rigid conformation of furan ring made **Int5** give **Z-6** stereoisomer as a

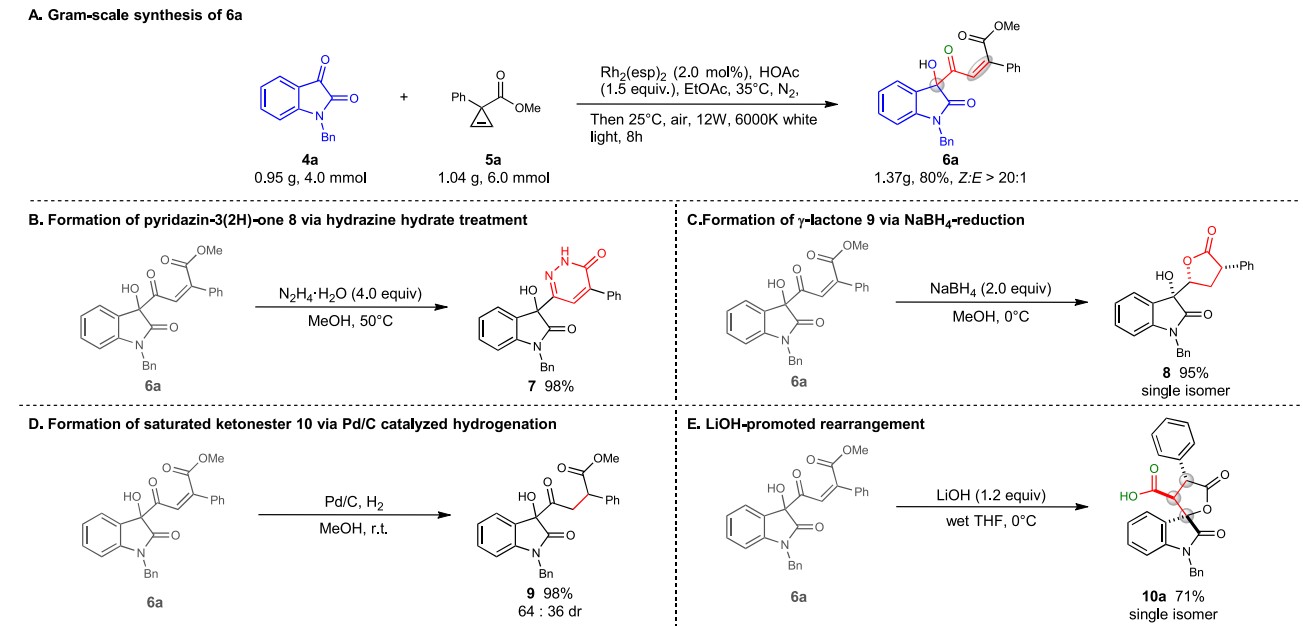

**Fig. 4 | Gram-scale PCR of 4a and 5a and synthetic utilities of PCR product 6a.**
**A** Gram-scale synthesis of **6a**. **B** Formation of pyridazin-3(2H)-one **8** via hydrazine hydrate treatment. **C** Formation of g-lactone **9** via NaBH$_4$-reduction. **D** Formation of saturated ketonester **10** via Pd/C catalyzed hydrogenation. **E** LiOH-promoted rearrangement.

kinetically favorable product; (ii) the **Z-6** stereoisomer was thermally stable than **E-6** stereoisomer by 18.56 kcal/mol. Last but not least, the control of diastereoselectivities of LiOH-promoted rearrangement was attributed to two key transition states, **TS4** and **TS5**. In two lithium cations-associated **TS4**, one of the lithium cations coordinated with two carbonyl groups of amide and ester, and the other one coordinated with the carbonyl group of ester and had cation-π interaction with the phenyl group. Through this interaction model, these two lithium cations locked **TS4** in the open chair-like conformation, which favored *syn*-product (Fig. 6B). The relative stereochemistry of the third stereocenter was set during protonation process, which went through **TS5**. In **TS5**, two lithium cations were bonding with four oxygen atoms and formed a boat-like conformation, allowing the lithium enolate to be protonated from the less hindered side to form *syn*-product.

Taking data together, we proposed a possible mechanism of PCR. As shown in Fig. 7, in the carbenoid transfer cycle, dirhodium tetraacetate catalyzed the cleavage of cyclopropene carboxylate **4** giving the dirhodium-associated carbenoid **Int1**. **Int1** underwent intramolecular attack on the carbenoid by carbonyl group of the ester and resulted in **Int2** equilibrating with **Int3**. The slightly more stable **Int3** participated in the followed aldol-type addition to isatin **5** and simultaneous aromatization of the adduct, affording compound **6a−Int** as an isolable intermediate. In the aerobic oxidation cycle, Rh$_2$(OAc)$_4$ was excited by visible light transfer energy to triplet oxygen and led to the generation of $^1$O$_2$. The $^1$O$_2$ underwent [3+2] cycloaddition with intermediate **6a−Int** to produce **Int4**. **Int4** was then reduced by HOAc to **Z-6** by going through **Int5**. Upon the treatment of LiOH, the hydroxy group of **Z-6** was deprotonated, and the resulting oxygen anion attacked the neighboring carbonyl group to form epoxide **Int7**. Subsequently, the opening of epoxide ring was followed by the negative charge transfer from oxygen to the 3-carbon of oxindole. The anion was stabilized by the aromatization resonance between **Int8** and **Int9**. The assumption of **Int8** or **Int9** was proved by the successful isolation of compound **12**, which could be further converted to the final product **10** by treating with LiOH in THF/H$_2$O (Fig. 5). Then **Int8** or **Int9** produced **Int11** via an intramolecular Michael-addition **TS4** was a possible key transition state of the conversion. **Int11** was then stereoselectively protonated to

produce **Int12**. The **Int12** gave **Int13** through intramolecular transesterification. Finally, the **Int13** was further protonated to produce product **10**.

## Biological function

To explore the biological function of the products of the PCR, we performed in silico screening against the protein targets of interest with Autodock Vina and found product **6** precisely docked into the pocket of protein tyrosine phosphatase 1B (PTP1B) (Supplementary Fig. 15). PTP1B is an enzyme which removes phosphate group from the tyrosine residues of its substrate protein. Inhibition of PTP1B compensates for insulin signaling pathway and enhances anti-tumor immunity. Therefore, searching for PTP1B inhibitors is of great significance in the treatment of type II diabetes and cancers[49–51]. We screened these compounds against PTP1B and TCPTP via 6,8-Difluoro-4-Methylumbelliferyl Phosphate assay. Preliminary experiment found that six compounds including **6a**, **6g**, **6i**, **6m**, **6n**, and **6r** inhibit PTP1B with $4.63 \pm 0.42$, $0.28 \pm 0.04$, $4.56 \pm 0.24$, $7.21 \pm 0.52$, $5.87 \pm 0.53$ and $3.75 \pm 0.39$ μM IC$_{50}$, and 2.76-, 8.14-, 1.26-, 0.51-, 6.84-, and 6.07-fold selectivity against its isoform TCPTP which shares 70% structural similarity, respectively (Supplementary Fig. 16). The results proved compounds **6g, 6n** and **6r** to be promising PTP1B inhibitors.

## Discussion

We discovered the undocumented bifunctionality of [Rh$_2$] in metalla-photocatalysis and developed a PCR catalyzed by the bifunctional [Rh$_2$]. Firstly, TD-DFT calculations on Rh$_2$(OAc)$_4$ and molecular orbital analysis indicated the [Rh$_2$] took uncommon MMCT excited states, thus differentiating themselves from the conventional MLCT photocatalysts. By harnessing the photocatalytic activity of the [Rh$_2$], carbenoid chemistry and $^1$O$_2$ chemistry were connected and an interesting PCR was developed based on the concept. The PCR is featured by high atom-efficiency, green conditions, easy purification, scalable synthesis and pharmaceutically interesting products. Furthermore, simple treatment of the products with LiOH led to structurally interesting γ-lactone spirooxindoles through an unreported rearrangement. DFT-aided mechanistic study revealed the PCR went

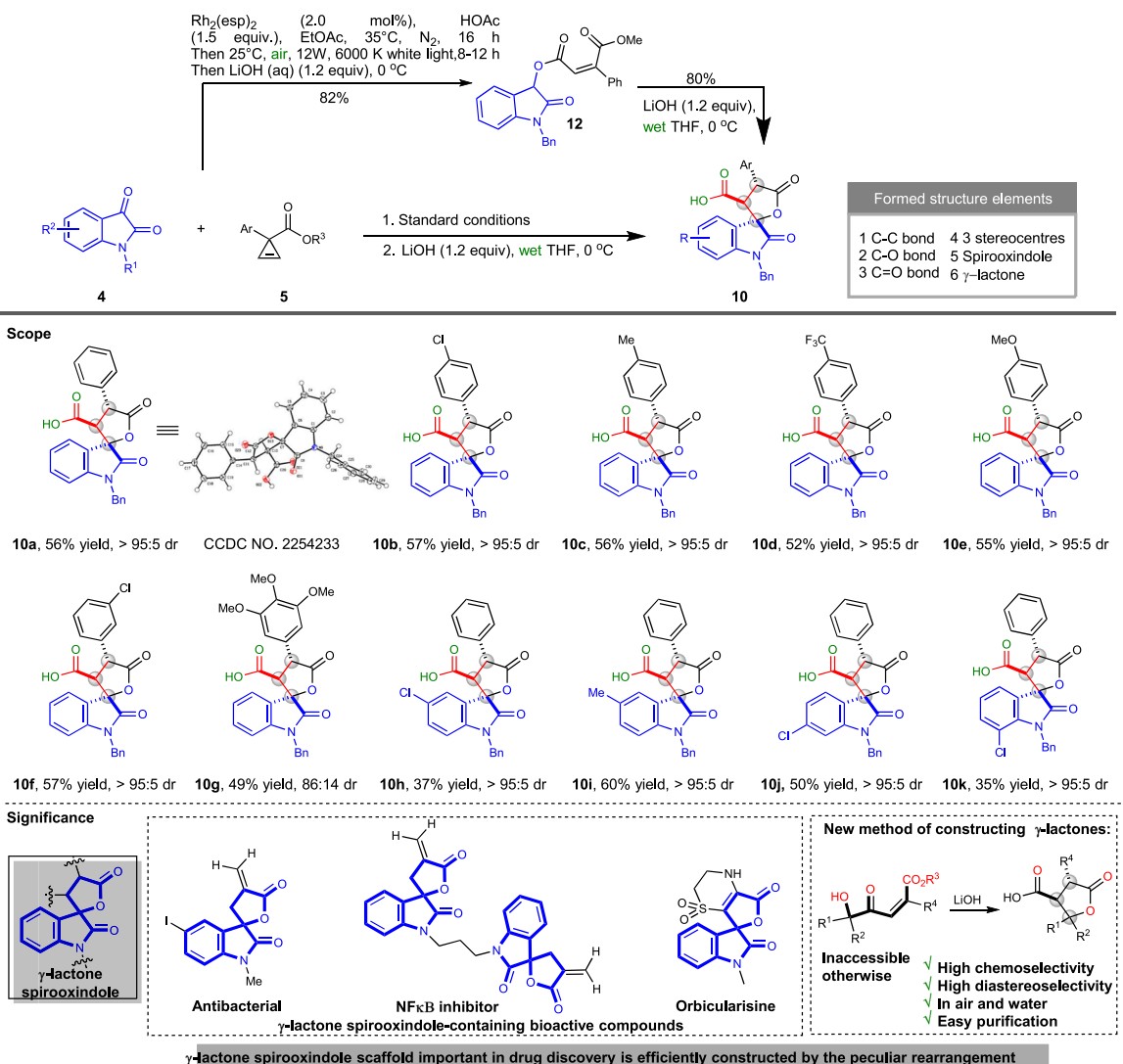

**Fig. 5 | Substrate scope and significance of Rh₂(esp)₂-catalyzed PCR of isatins and cyclopropenes.** Reaction conditions: **4a** (0.20 mmol), **5a** (0.30 mmol), Rh₂(esp)₂ (2.0 mol%), ethyl acetate (2 mL), sunlight, then switched solvent to THF and added 1.2 equiv. LiOH (aq).

through a zwitterion **Int3** and interpreted the origin of the Z-selectivity of **6** and the stereoselectivity of **10**. Last but not least, the products of the reaction showed inhibitory activity against PTP1B, paving the way for the discovery of PTP1B-based anti-diabetic and anti-cancer drugs. We believe this work will shed light on discovering bifunctional catalysts for metallaphotocatalysis and on developing PCRs for exploring chemical space. Development of new [Rh₂]-catalyzed PCRs and further hit-to-lead optimization of the PTP1B inhibitors is ongoing in our lab and the results will be reported shortly.

## Methods
### Representative procedure of dirhodium tetracarboxylates-catalyzed PCR
An oven-dried 5 ml Schlenk tube was charged with isatin **4** (1.0 equiv.), Rh₂(esp)₂ (2 mol%), acetic acid (1.5 equiv.) and EtOAc (0.1 M). After the mixture was thoroughly degassed and filled with nitrogen, cycloprop-2-ene-1-ester **5** (1.5 equiv.) was added and the Schlenk tube was tightly sealed. The reaction was stirred at 35 °C under nitrogen atmosphere for 16−24 h (monitored by thin-layer chromatography), then the reaction was stirred at 25 °C under air atmosphere and visible light irradiation (12 W white LEDs or sunlight) for 8−12 h. After complete consumption of the reaction intermediate **6**−**Int** monitored by TLC, the reaction mixture was filtered, concentrated under vacuum. The

residue was purified by silica gel flash column chromatography (petroleum ether/EtOAc = 10/1-3/1) to afford product **6**.

### Representative procedure for PCR and LiOH-promoted rearrangement transformations in a sequential process
An oven-dried 5 ml Schlenk tube was charged with isatin **4** (1.0 equiv.), Rh₂(esp)₂ (2 mol%), acetic acid (1.5 equiv.) and EtOAc (0.1 M). After the mixture was thoroughly degassed and filled with nitrogen, cycloprop-2-ene-1-ester **5** (1.5 equiv.) was added and the Schlenk tube was tightly sealed. The reaction was stirred at 35 °C under nitrogen atmosphere for 16−24 h (monitored by TLC), then the reaction was stirred at 25 °C under air atmosphere and visible light irradiation (12 W white LEDs or sunlight) for 8−12 h. After complete consumption of the reaction intermediate **6**−**Int** monitored by TLC, the reaction mixture was filtered and concentrated under vacuum. In a clean round-bottomed flask, the residue and THF (0.1 M) were added. The mixture was cooled in an ice bath and then a solution of 1.2 M aqueous LiOH (1.2 equiv.) was added slowly. The reaction system stirred at 0 °C for 15-40 min. After completion, 1 M HCl was added to render the solution acidic (pH = 1-2), and the mixture was extracted three times with EtOAc, washed with brine, dried over Na₂SO₄, filtered and concentrated in a vacuum. The crude residue was purified by silica gel flash column chromatography (CH₂Cl₂/MeOH = 50/1-10/1) to afford product **10**.

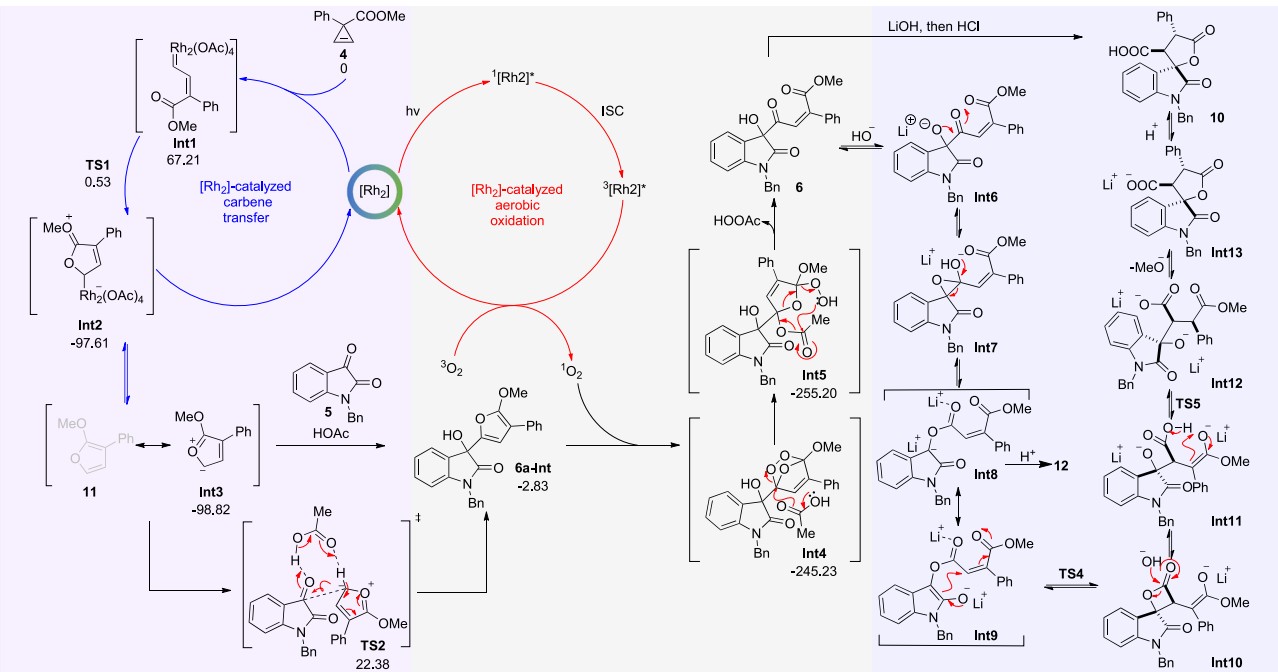

**A. Gibbs energy map of [Rh₂]-catalyzed PCR of 4a and 5a.**

**B. Key transitional states of LiOH-promoted rearrangement of 6a.**

**Fig. 6 | Mechanism investigation via DFT calculations on M062X/6-31G(d). A** Gibbs energy map of the PCR of **4a** and **5a**. **B** Key transitional states of LiOH-promoted rearrangement of **6a**.

**Fig. 7 | Proposed mechanism of [Rh₂]-catalyzed PCR and LiOH-promoted rearrangement of 6a.** ISC: intersystem crossing.

## Reporting summary

Further information on research design is available in the Nature Portfolio Reporting Summary linked to this article.

## Data availability

Materials and methods, experimental procedures, mechanistic studies, computational studies, bioactivity assessment and NMR spectra are provided in the Supplementary Information. *In virtual* library of the related compounds can be downloaded free of charge via http://www.sysu-sps-compound.com/ui/#/molecule-library/29. Source data of the coordinates of the optimized structures are provided (cartesian coordinates.xlsx). Crystallographic data for the structures reported in this article have been deposited at the Cambridge Crystallographic Data Centre, under deposition numbers CCDC 2249289[6a], 2254233[10a]. Copies of the data can be obtained free of charge via https://www.ccdc.cam.ac.uk/structures. PTP1B co-crystal structure was freely downloaded from https://www.rcsb.org/ (PDBID: 1T4J). All other data are available from the corresponding author upon request. Source Data are provided with this paper.

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

## Acknowledgements

We are grateful for the support by the National Natural Science Foundation of China (Nos. 92056201 (W.H.), 82003592 (T.S.)). Fundamental Research Funds for the Central Universities, Sun Yat-sen University (No. 23ptpy167). Key-Area Research and Development Program of Guangdong Province (No. 2022B1111050003) (W.H.).

## Author contributions

T.S. contributed to the concept, manuscript preparation, computational study, instructions on experiments, and Supplementary Information. T.Z. contributed to chemistry experiments and Supplementary Information. J.Y. contributed to bioactivity evaluation. Y.L. contributed to substrate preparation. J.S. contributed to construction of in silico library. J.Z. contributed to bioactivity evaluation. M.Z. contributed to substrate preparation. D.Z. contributed to concept. W.H. contributed to concept and project management.

## Competing interests
The authors declare no competing interests.
