## [Peer Review File · Nature Communications]

Bifunctionality of Dirhodium Tetracarboxylates in MetallaphotocatalysisReviewers' Comments:

Reviewer #1:

Remarks to the Author:

The authors demonstrated an attractive metallaphotocatalysis with a bifunctional dirhodium tetracarboxylate catalyst, showing an unprecedented photocatalytic activity via a rarely used metal-to-metal charge transfer transition in photocatalysis. A photochemical cascade reaction was realized via a combination of carbene chemistry and singlet oxygen chemistry by Rh₂. Various valuable products were synthesized in good results with high atom-efficiency and excellent stereoselectivities under mild conditions. Besides, rationalizing the reaction pathway and interpreting stereoselectivities of the reaction were proposed based on DFT calculations. Overall, this manuscript is recommended for publication after addressing the following comments.

1. I am interested in the absorption band of charge transition between Rh-Rh δ and Rh-Rh σ^* , since it should be located between 387nm to 984nm according the DFT calculation in the manuscript. However, no absorption band of this charge transition was observed on the UV-vis spectrum (Fig. 2A). In addition, according to the Laporte selection rule, the transition from molecular orbitals $u \rightarrow u$ is forbidden. However, both n to σ^* and δ^* to σ^* in Fig 2B are $u \rightarrow u$ transitions. So it is suggested the authors to reconsider the analysis of the UV- spectrum.
2. According to entry 1 of Table 1, the reaction occurred smoothly in the absence of HOAc. In this case, how is Int4 converted into the product (Scheme 2)?
3. In Eq. 1 of Figure S3, when the reaction was conducted in the dark, product 6a was still detected in 22% yield. It seems impossible because a photochemical process was proposed to be involved in the catalysis. How to explain this result?
4. In Eq. 3 of Figure S3, when chiral dirhodium complexes were used, the ee values of product 6a were 0. It is suggested to provide the yield of 6a in each cases.
5. If the PCR reaction is conducted in air under visible light in the whole process, what will be observed?
6. Can the by-product S4 be further oxidized to 6a in the presence of singlet oxygen?
7. How about the substrate scopes of dicarboxylate cyclopropenes or alkyl-carboxylate cyclopropenes? If the results are negative, they should be mentioned and added in the SI.
8. There are several minor errors in the manuscript. For example, in Fig 1E, the structure of the rhodium carbene intermediate is incorrect. Line 10, the word "unprecedented" should be "unprecedented". Please check the whole manuscript carefully.

Reviewer #2:

Remarks to the Author:

In this paper, Profs. Shi and Hu present a photochemical cascade reaction enabled by a bifunctional dirhodium catalyst. In the reaction, Rh₂(esp)₂ serves as both a transition-metal catalyst and a photosensitizer to connect carbene transfer and singlet oxygen oxidation. Interestingly, a metal-to-metal charge transfer transition is proposed to be in operation. A series of experiments were performed to reveal the photocatalytic activity of the rhodium catalyst and to support the proposed mechanism. Overall, these experiments and conclusions obtained thereby are reasonable to this reviewer.

Moreover, the substrate scope, synthetic transformations, and the access to biological-active compounds demonstrate the potential utility of this method.

In conclusion, this is a nice report with high novelty and broad synthetic utility. Therefore, I strongly recommend the acceptance of this work after minor modifications.

1. The key selling point is the bifunctionality of the rhodium complex. However, the authors ignored the work by Meggers on bis-cyclometalated rhodium complexes, which in many cases can act as both chiral Lewis acid photosensitizer (J. Am. Chem. Soc. 2017, 139 (27), 9120-9123) or photoredox catalyst (Angew. Chem. Int. Ed. Engl. 2018, 57 (19), 5454-5458.). Also see their review paper, Acc. Chem. Res. 2019, 52 (3), 833-847. Meggers' work might be the only precedence prior to this work on

using rhodium complexes as photocatalysts. It deserves to be highlighted in the introduction.

2. Partly out of the above reason, the title could be revised too. In addition, for a Nat Commun paper, the title does not have to be so attention-grabbing.

3. Please rephrase the sentence on Page 1 lines 24-26: "Glorious (7), Gevorgyan (8), ... have developed photosensitizer tethered bifunctional catalysts which ..." No all the multifunctional chiral photocatalysts consist of the tethered chromophore and catalytic center.

4. What is the excited Rh-species? The authors state that it is diRh-tetraacetate. How to exclude the possibility of the rhodium-carbene (Int1 in Scheme 2). Additionally, the calculated and experimental spectra differed too much. More control experiments on UV-Vis absorption and trapping experiments of singlet oxygen may be helpful to identify the excited Rh-species.

5. The reaction uses sunlight or white LED as the light source. Given the issue of reproduction, showing the best wavelength and light intensity may be helpful.

Other minor issues:

1. Page 5 line 202, change 'triple' to "triplet"

2. Some compounds, such as 9, 10g, may need further purification.

3. The compound numbers in the legend of Scheme 1 are inconsistent with the content.

4. In the procedure for the synthesis of compound 10, the reaction mixture needs to be worked up and transferred to another round bottom flask, then other reagents and solvent need to be added. This is not strictly a one-pot synthesis.

Dear reviewers,

We are very grateful for your insightful and constructive suggestions on this manuscript. Before the presentation of our point-by-point response to your comments, please allow us to explain to you two things. Firstly, the experiment for the revision was interrupted by Long Covid. Fortunately, the authors bounced back and made the revision. We hope the lagging response doesn't consume your patience and interest to the manuscript. Additionally, after reevaluation of the contribution of authors in this manuscript (The detailed assignment of contribution is presented in the manuscript), we flipped the order of the two first-coauthors. We are truly sorry for the inconvenience caused to you. We believe the quality of the manuscript improved after the revision guided by you and it is suitable for the publication in Nature Communications.

Review 1.

1. I am interested in the absorption band of charge transition between Rh-Rh δ and Rh-Rh σ^* , since it should be located between 387nm to 984nm according to the DFT calculation in the manuscript. However, no absorption band of this charge transition was observed on the UV-vis spectrum (Fig. 2A). In addition, according to the Laporte selection rule, the transition from molecular orbitals $u \rightarrow u$ is forbidden. However, both π to σ^* and δ^* to σ^* in Fig 2B are $u \rightarrow u$ transitions. So it is suggested the authors to reconsider the analysis of the UV- spectrum.

Thank you for pointing out the difference of UV-vis spectrum between calculational and experimental version. We previously conducted TD-DFT calculation without considering the coordination effect of ethyl acetate. The ignoration of the perturbation of the Rh-Rh $\pi^* \rightarrow \sigma^*$ HOMO-LUMO transition by weak EtOAc coordination leads to the red-shift (*Inorg. Chem.* **2015**, *54*, 8817–8824), comparing to the experimental spectrum. And the wrongly predicted absorption peaks led to the incorrect assignment of charge transition. Therefore, we

reconducted geometry optimization and TD-DFT by considering coordination effect of ethyl acetate. As shown in the right box, the calculational and experimental spectra are consistent and the assignment of charge transition obeys to the Laporte selection rule.

2. according to entry 1 of Table 1, the reaction occurred smoothly in the absence of HOAc. In this case, how is Int4 converted into the product (Scheme 2)?

According to the references, hydrolysis of endopeptide intermediate followed by elimination of the hydrogen peroxide anion yields the 4-oxo-2-enal. In the case of conversion of Int4 into the product Z-6, no product was observed under anhydrous condition. Based on the known case and control experiment, we proposed the mechanism in the following figure.

3. Eq.1 of Figure S3, when the reaction was conducted in the dark, product 6a was still detected in 22% yield. It seems impossible because a photochemical process was proposed to be involved in the catalysis. How to explain this result?

Thank you for pointing out the contradiction. We redid the control experiment in more strict photon-free condition and it turned out no desired PCR product was observed. We postulate that some photons still penetrated into the reaction tube in our previous control experiment.

4. Eq. 3 of Figure S3, when chiral dirhodium complexes were used, the ee values of product 6a were 0. It is suggested to provide the yield of 6a in each cases.

Thank you for your suggestion! The yields of 6a in each case has been added in the Eq. 3 of Figure S3.

5. If the PCR reaction is conducted in air under visible light in the whole process, what will be observed?

The desired product Z-6 was still harvested in 52% yield, and the major by-product was the 4-oxo-2-enal which generated from carbene transfer/zwitterion formation/ 1O_2 oxidation of cyclopropene.

1H NMR (400 MHz, Chloroform- d)

6. Can the by-product S4 be further oxidized to 6a in the presence of singlet oxygen?

This is an interesting question! S4 was tested under standard oxidation condition, but no new product was observed.

7. How about the substrate scopes of dicarboxylate cyclopropenes or alkyl-carboxylate cyclopropenes? If the results are negative, they should be mentioned and added in the SI.

Thank you for the informative suggestion. We found the reaction with ethyl dicarboxylate cyclopropene could afford the furan intermediate, but it failed to be oxidized to the corresponding 1,4-dicarbonyl *Z*-alkene. We speculate the electron-deficient carboxyl group lowers the electron density of the furan ring and makes it harder to oxidize. Upon alkyl-carboxylate cyclopropene, we didn't detect the desired benzyl-carboxylate cyclopropene according to the reported procedure but observed the furan product (*ACS Catal.* **2021**, *11*, 10789-10795). With the information in mind, we tried to generate the unstable benzyl-carboxylate cyclopropene in situ by running a three-component reaction of benzyldiazoacetate, alkyne, and isatin. However, what we isolated is the product of the three-component reaction of benzyldiazoacetate, water, and isatin (*Green Chem.* **2003**, *15*, 620-624). Compared to its aryl-counterparts, we assume alkyl-carboxylate cyclopropenes losing the stabilization of carbene or zwitterion intermediates by the resonance with aryl

groups become less stable and can't be captured by the isatin.

Furan is detected by crude NMR of the mixture of the reaction, similar reaction catalyzed by Pd see: *ACS Catal.* **2021**, *11*, 10789–10795

ref.: *J. Am. Chem. Soc.* **2017**, *139*, 8364–837

Mechanism see: *Green Chem.*, **2013**, *15*, 620-624

¹H NMR (400 MHz, Chloroform-d)

¹H NMR (400 MHz, Chloroform-d)

8. There are several minor errors in the manuscript. For example, in Fig 1E, the structure of the rhodium carbene intermediate is incorrect. Line 10, the word “unprecedented” should be “unprecedented”. Please check the whole manuscript carefully.

Thank you for your kind and patient corrections. We have revised these mistakes and checked the whole manuscript carefully.

Review 2

1. The key selling point is the bifunctionality of the rhodium complex. However, the authors ignored the work by Meggers on bis-cyclometalated rhodium complexes, which in many cases can act as both chiral Lewis acid photosensitizer (J. Am. Chem. Soc. 2017, 139 (27), 9120-9123) or photoredox catalyst (Angew. Chem. Int. Ed. Engl. 2018, 57 (19), 5454-5458). Also see their review paper, Acc. Chem. Res. 2019, 52 (3), 833-847. Meggers' work might

be the only precedence prior to this work on using rhodium complexes as photocatalysts. It deserves to be highlighted in the introduction.

Thank you for pointing out the pioneering work of Meggers on bifunctional bis-cyclometalated rhodium complexes. We have highlighted this work in the introduction.

- Partly out of the above reason, the title could be revised too. In addition, for a Nat Commun paper, the title does not have to be so attention-grabbing.

According to your advice, we changed the title to “Unprecedented Bifunctionality of Dirhodium Tetracarboxylates in Metallaphotocatalysis”.

- Please rephrase the sentence on Page 1 lines 24-26: “Glorious (7), Gevorgyan (8), ... have developed photosensitizer tethered bifunctional catalysts which ...” No all the multifunctional chiral photocatalysts consist of the tethered chromophore and catalytic center.

Thank you for bringing this up! As you say, some are non-tethered bifunctional catalysts. So we deleted the phrase “photosensitizer tethered”.

- What is the excited Rh-species? The authors state that it is diRh-tetraacetate. How to exclude the possibility of the rhodium-carbene (Int1 in Scheme 2). Additionally, the calculated and experimental spectra differed too much. More control experiments on UV-Vis absorption and trapping experiments of singlet oxygen may be helpful to identify the excited Rh-species.

Thank you for your interesting proposal about the excited Rh-species. We isolated the intermediate **7** and conducted $^1\text{O}_2$ oxidation under the standard condition, and **7** smoothly converted into **Z-6**, indicating $[\text{Rh}_2]$ is the excited species (Supplementary Fig.3, eq.1). However, we could not exclude that the $[\text{Rh}_2]$ -**7a** complex is the excited Rh-species. To explore this, we trace the oxidation process with UV-vis and no obvious UV-vis spectrum shift was observed during the whole process, implicating the $[\text{Rh}_2]$ itself is the excited species.

Additionally, the TD-DFT calculations were redone by considering the coordination effect of the EtOAc. And the revised calculated spectra is consistent with the experimental spectra (For details see Fig. 2 in manuscript).

^aThe yields were determined by crude ^1H NMR using 1,3,5-trimethoxybenzene as the internal standard. The ee value was determined by HPLC analysis using a chiral stationary phase. ^bIsolated yield. ^cCondition A: standard conditions; Condition B: dark reaction, 48h; Condition C: no $\text{Rh}_2(\text{esp})_2$, 48h; Condition D: N_2 instead of O_2 .

5. The reaction uses sunlight or white LED as the light source. Given the issue of reproduction, showing the best wavelength and light intensity may be helpful.

Thank you for your kind suggestions. We have added the parameters of the light used in the optimized conditions.

6. Page 5 line 202, change 'triple' to "triplet"

Thank you for your kind correction! We have revised and highlighted with yellow in the MS.

7. Some compounds, such as **9**, **10g**, may need further purification

Thank you for your helpful feedback, we have repurified **10g** by separating the two diastereoisomers of **10g**. However, we failed many times with different separation methods to separate the diastereoisomers of **9**. In light of the difficulty, we ask for permission to present **9** as the mixture of diastereoisomers.

8. The compound numbers in the legend of Scheme 1 are inconsistent with the content

Thank you for your kind correction! We have revised and highlighted with yellow in the MS.

9. In the procedure for the synthesis of compound 10, the reaction mixture needs to be worked up and transferred to another round bottom flask, then other reagents and solvent need to be added. This is not strictly a one-pot synthesis.

Thank you for your correction! We have replaced “one-pot” with a “sequential process”. Inspired your comment on one-pot synthesis, we conducted a strictly one-pot synthesis. Interestingly, we obtained the intermediate **12** which could convert into the final product **10**. The observation not only prove a one-pot procedure to compound **12** is feasible but also support our proposed mechanism of hydrolysis and rearrangement to spirooxindole **10**.

Overall, thank you again for reviewers' helpful comments. We think the current manuscript is suitable to publish in *Nature Communications*.

Yours sincerely

Taoda Shi and Wenhao Hu

史滔达 胡皓

Reviewers' Comments:

Reviewer #1:

Remarks to the Author:

All the comments have been addressed. The manuscript can be published as it is.

Reviewer #2:

Remarks to the Author:

The authors have addressed my comments as well as those of the other reviewer. I would recommend the publication of this paper at this stage. Congratulations!